# Enhanced joint impact of western hemispheric precursors increases extreme El Niño frequency under greenhouse warming

Hyun-Su Jo [1] & Yoo-Geun Ham [1] ✉

Sea surface temperature variability over the north tropical Atlantic (NTA) and over the subtropical northeast Pacific (SNP), which is referred to as the North Pacific Meridional Mode, during the early boreal spring is known to trigger El Niño–Southern Oscillation (ENSO) events. The future changes of the influence of those northwestern hemispheric precursors on ENSO are usually examined separately, even though their joint impacts significantly differ from the individual impacts. Here, we show that the impacts of both NTA and SNP on ENSO significantly increase under greenhouse warming and that the degrees of enhancement are closely linked. The wetter mean state over the off-equatorial eastern Pacific is a single contributor that controls the impacts of both NTA and SNP on ENSO. The enhanced joint impacts of the northwestern hemispheric precursors on ENSO increase the occurrences of extreme El Niño events and the ENSO predictability under greenhouse warming.

The El Niño–Southern Oscillation (ENSO), the irregular fluctuation between the warm El Niño and cold La Niña conditions over the tropical Pacific, stems from interannual climate variability that affects weather, ecosystems, and food production worldwide[1–4]. The lead–lag relationship between the warm water volume (WWV) over the equatorial Pacific and sea surface temperature (SST) associated with ENSO has significantly weakened since the 2000s[5], signifying that the WWV-related recharge/discharge process driving the transition of the ENSO phase has become less efficient in recent decades. In contrast, the influence of SST variabilities from outside the tropical Pacific on ENSO, particularly over the North Tropical Atlantic (NTA)[6] or the subtropical northeast Pacific (SNP)[7], referred to as the North Pacific Meridional Mode (NPMM)[8], has considerably increased in recent decades[9–12].

Whether the recently observed enhancement of the impact of SST anomalies over NTA and SNP on ENSO stems from global warming needs to be determined[13–17]. Global climate models generally simulate the enhanced impact of NPMM on ENSO in the future climate[11,16]. They have shown that both wind-evaporation-SST (WES) feedback and atmospheric convection response to the SST anomalies over the subtropical North Pacific operate more effectively in a background of higher SSTs in the future climate.

However, whether the impact of NTA SST on ENSO increases or decreases under greenhouse warming remains controversial. Previous studies have reported a decrease in the future climate using an intermediate climate model consisting of a full atmospheric GCM coupled with a reduced-gravity ocean (RGO) model and the Coupled Model Intercomparison Project's Phase 5 (CMIP5) models, which contrasts with recent observational changes[13,14]. The weakening of the impact of NTA SST on ENSO has been attributed to the weak coupled atmosphere-ocean feedbacks within the Pacific[18]. However, the NTA–ENSO coupling is expected to be enhanced due to the intensification of ENSO-induced atmospheric responses over the tropical Pacific under greenhouse warming[19,20]; moreover, the NTA variability and occurrences of extreme NTA events are projected to be intensified under greenhouse warming[17].

The impact of NTA and SNP SST on ENSO has been separately assessed, despite their similar physical mechanisms. The SNP-related SST and atmospheric anomalies over the subtropical central–eastern Pacific propagate southwestward to the tropical western–central Pacific via the low-level meridional wind-involved WES feedback[12]. Similarly, the NTA-induced signals over the subtropical central–eastern Pacific propagate to the tropical Pacific through the equatorward extension of the SST and precipitation anomalies driven

[1]Department of Oceanography, Chonnam National University, Gwangju, South Korea. ✉e-mail: ygham@chonnam.ac.kr

by the low-level meridional wind[21,22]. Given the similarities in physical mechanisms, a single aspect is likely to exist that controls the change in the impacts of both NTA and SNP SST on ENSO. Using the outputs from the CMIP5 and the Coupled Model Intercomparison Project's Phase 6 (CMIP6) models, this study shows that a wetter mean state over the off-equatorial eastern Pacific is the single key constraint that is responsible for enhancing the impact of the northwestern hemispheric precursors (i.e., NTA and SNP SST) on ENSO under greenhouse warming.

## Results

### Joint enhancement of the impact of ENSO precursors due to global warming

The ability of the CMIP5 and CMIP6 models to reproduce the observed NTA- and SNP-related responses in historical simulations over the 1951–1999 period (referred to as the present-day climate) is assessed. The NTA and SNP indices during February–March–April (FMA0, where "0" denotes the current year) are defined by taking the area-averaged SST anomalies that are quadratically detrended over NTA (80° W–20° E, 0°–15° N) and SNP (170°–120° W, 5°–25° N), respectively. The degree of impact of NTA and SNP SST is quantified using the partial lagged regression coefficient of the quadratically detrended equatorial Pacific SST anomalies (140° E–80° W, 5° S–5° N)

during the following December–January–February (D0JF1, where "1" denotes the following year) onto the respective FMA0 NTA and SNP indices after removing the ENSO signals (see Methods for details of the partial regression) (Fig. 1a, b). Note that the regressed coefficients for NTA SST are multiplied by −1 to facilitate comparison with the regressed coefficients for SNP SST.

In total, 47 models (17 CMIP5 and 30 CMIP6 models, corresponding to approximately 54% of all CMIP models) that simulate a warming response over the tropical Pacific to negative NTA and positive SNP SST in the present-day climate are selected. In the CMIP5 and CMIP6 models, the multi-model averaged regression coefficients are 0.14 ± 0.10°C·s.d.$^{-1}$ and 0.17 ± 0.10°C·s.d.$^{-1}$ for NTA and 0.20 ± 0.12°C·s.d.$^{-1}$ and 0.29 ± 0.13°C·s.d.$^{-1}$ for SNP, respectively, rejecting the null hypothesis that the regression coefficient is zero. Furthermore, the observed regression coefficients (i.e., 0.24°C·s.d.$^{-1}$ for NTA and 0.26°C·s.d.$^{-1}$ for SNP) are within the 95% confidence range of the multi-model averaged regression coefficient, indicating that the selected climate models realistically simulate the lag relationship between NTA or SNP SST and ENSO to some extent in the present-day climate. It should be noted that the CMIP6 models accurately simulate the phase-locking of ENSO, NTA, and SNP as the observed to some extent, while the simulation of the NTA seasonal cycle is relatively poor in the CMIP5

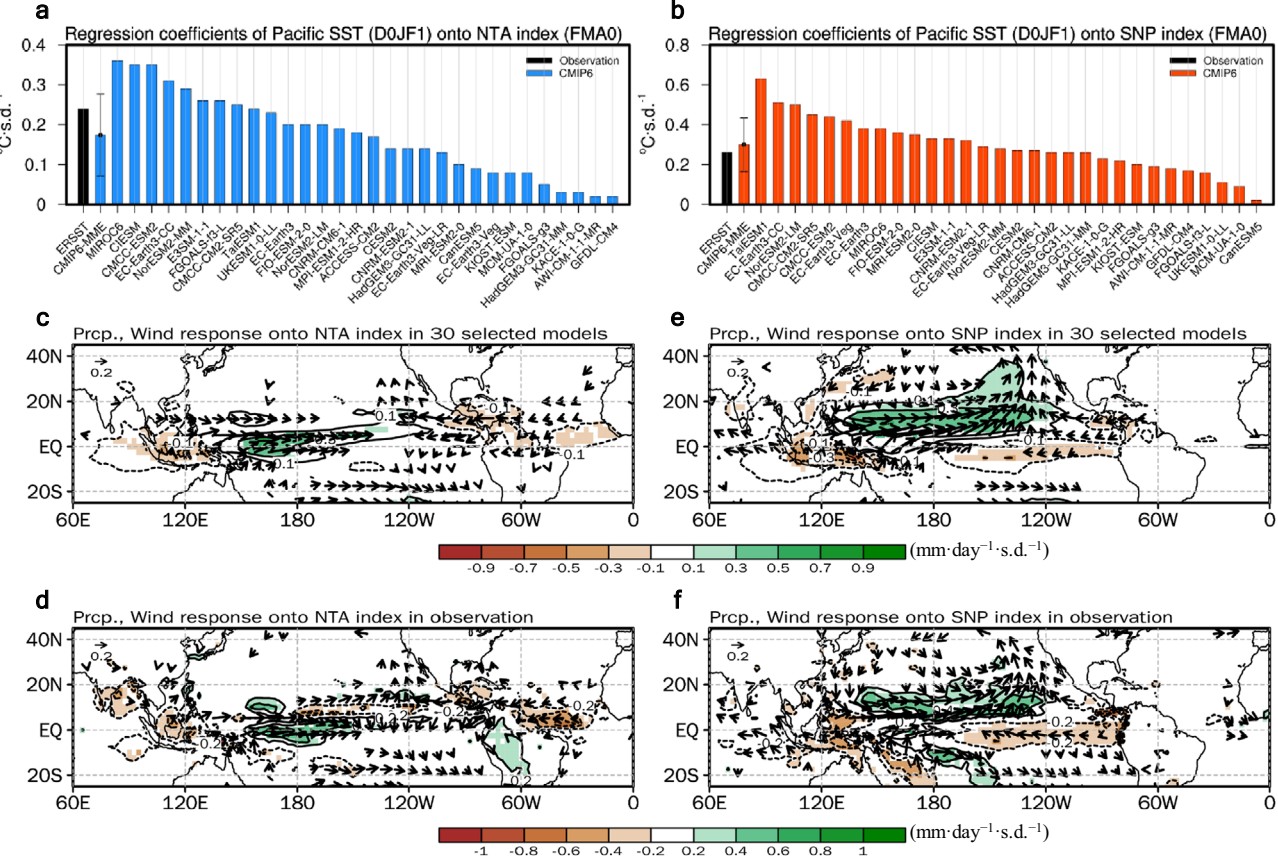

**Fig. 1 | Observed and modeled impact of northwestern hemispheric precursors on El Niño-Southern Oscillation (ENSO). a** Equatorial Pacific sea surface temperature (SST) response (°C·s.d.$^{-1}$) to the north tropical Atlantic (NTA) SST during the period 1951–2020 based on observations (black bar) and the period 1951–1999 based on the CMIP6 models (blue bars), which is measured as the area-averaged regression coefficients of grid-point equatorial Pacific SST anomalies during the following winter (D0JF1) (140° E–80° W, 5° S–5° N) onto the NTA SST during the early spring (FMA0) after removing the linearly regressed cold tongue index (CTI) during the preceding winter. **b** Same as **a** but for the subtropical northeast Pacific (SNP) SST after removing the CTI during FMA0 (red bars). **c**–**f** Spatial patterns of the Pacific response shown as the regression of precipitation (mm·day$^{-1}$·s.d.$^{-1}$;

contours and shading) and 925 hPa wind (10 m wind in observations) (m·s$^{-1}$·s.d.$^{-1}$; vectors) anomalies during the following spring to fall seasons (February–October; FMAMJJASO0) onto the FMA0 NTA and SNP SST based on the multi-model averaged value of the 30 selected CMIP6 models (**c**, **e**) and observations (**d**, **f**) under the present-day climate. Error bars in **a**, **b** represent the multi-model s.d. of the 30 selected models. Shading and vectors in **d**, **f** represent features that exceed the 95% confidence level or the most robust ensemble features where the mean exceeds 1.0 s.d. in **c**,**e** are shown. The color bars in **c**–**f** represent the regression coefficients (mm·day$^{-1}$·s.d.$^{-1}$). The regressed coefficients are multiplied by −1 for the NTA SST.

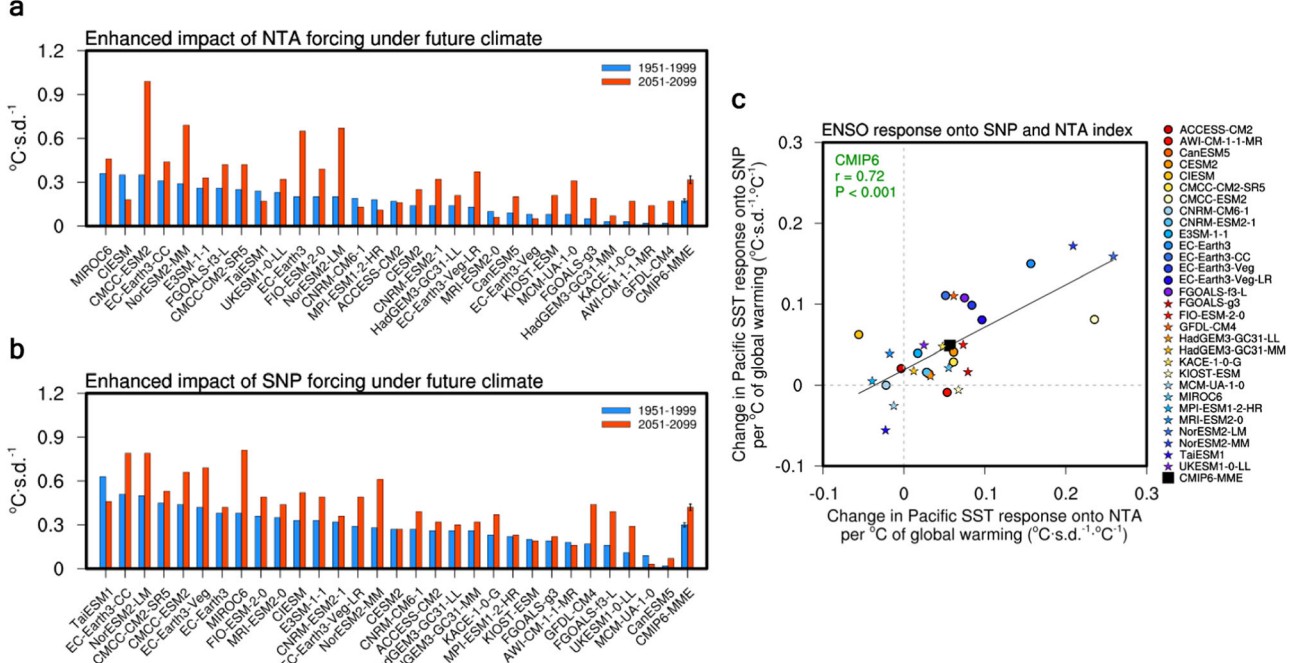

**Fig. 2 | Enhanced joint impacts of north-western hemispheric precursors on El Niño-Southern Oscillation (ENSO) under greenhouse warming. a** Comparison of the DOJF1 equatorial Pacific SST response (°C·s.d.⁻¹) to the FMA0 north tropical Atlantic (NTA) SST under the present-day climate (1951–1999; blue bars) and future climate (2051–2099; red bars) in the 30 selected CMIP6 models. The multi-model averaged value and corresponding error bars for the 30 selected CMIP6 models (labeled CMIP6-MME) is also shown. **b** Same as **a** but for the FMA0 subtropical northeast Pacific (SNP) SST. Error bars are calculated as 1.0 s.d. of 10,000 inter-realizations of a bootstrap method (see Methods for details on the bootstrap test). **c** Inter-model relationship between changes (future minus present day) of the DOJF1 equatorial Pacific SST response to the respective FMA0 SNP SST (*y* axis; °C·s.d.⁻¹·°C⁻¹) and FMA0 NTA SST (*x* axis; °C·s.d.⁻¹·°C⁻¹). For the models with available data, the CMIP6 models are denoted by dots and stars (see CMIP6 data in Methods). A linear fit is displayed along with the correlation coefficient *r* and *P* value based on the 30 selected models (black line and green text). The multi-model averaged value for the CMIP6 models (labeled CMIP6-MME) is also denoted by black square. The changes in each model are scaled by the corresponding increase in the global mean temperature. The regressed coefficients are multiplied by −1 for the NTA SST.

models (Figs. S1 and S2). Therefore, we focused on the CMIP6 model results to assess the changes in the role of NTA or SNP on the ENSO due to the global warming.

The overall spatial distributions of NTA-cooling- or SNP-warming-induced SST anomalies during the simultaneous (FMA0) and following seasons (May–June–July (MJJ0) and August–September–October (ASO0)) are also realistically simulated in the 30 selected CMIP6 models (Figs. S3 and S4). The results demonstrate that the modeled spatial patterns of precipitation and low-level wind anomalies during the subsequent spring-to-fall seasons of NTA-cooling and SNP-warming are generally comparable to those observed[11,23] (Fig. 1c–f). In detail, NTA-cooling during FMA0 reduces in-situ convective activities during the following seasons over the off-equatorial northern Atlantic and the Atlantic warm pool (Fig. 1c, d). This is responsible for the westerly anomalies over the equatorial western–central Pacific and the southeasterly anomalies over the off-equatorial eastern Pacific, which consequently increase in-situ convective anomalies in both observations and climate models, respectively[6,23]. The associated dipole wind responses over the equatorial Pacific (i.e., westerlies and easterlies over the equatorial central and eastern Pacific, respectively) induce positive SST anomalies primarily over the equatorial central Pacific (Fig. S3).

In terms of the SNP-warming, the positive SST anomalies during FMA0 induce low-level cyclonic circulations over the subtropical central–eastern Pacific, weakening of the trade winds (Fig. 1e, f). The associated decrease in the wind speed reduces the evaporative heat loss and subsequently warms the SST. The warm SST anomalies propagate southwestward toward the tropical western–central Pacific via the WES feedback[24] (Fig. S4). Such atmosphere–ocean coupled

processes lead to the growth and equatorward propagation of the SNP-related SST and low-level wind anomalies during the subsequent seasons, which lead to ENSO events in the following winter. Those impacts of both NTA and SNP on the subsequent ENSO are confirmed by a set of partially-coupled model experiments by imposing NTA and SNP SST anomaly forcing (see Methods for details) (Fig. S5), which would reject the possibility that the NTA-, and SNP-related ENSO is a statistical artifact due to the oscillatory feature of the ENSO[25].

The spatial distributions of NTA-cooling- and SNP-warming-related atmospheric anomalies are similar, particularly over the off-equatorial central–eastern Pacific (Fig. 1c–f). For both NTA and SNP, positive precipitation anomalies and the southwesterly anomalies are prominent over the region. This suggests that the atmospheric responses over the off-equatorial central–eastern Pacific play an important role in transmitting the remote NTA and SNP SST signals to the equatorial Pacific[6,8].

Next, the changes in the impacts of NTA and SNP SST on ENSO under greenhouse warming is examined. Partial lagged regressions of the detrended DOJF1 equatorial Pacific SST anomalies onto the FMA0 NTA and SNP indices during the future period (2051–2099) are enhanced compared to those during the present-day period (1951–1999) for a total of 23 and 26 of the 30 selected CMIP6 models (76.6% and 86.6%), respectively (Fig. 2a, b). The multi-model averaged value in the future climate also increases by 76.4% and 41.3% for NTA and SNP compared to that in the present-day climate, respectively. This increase is statistically significant at the 95% confidence level using the bootstrap method (see Methods for details on the bootstrap test). This increase is still robust and significant when the 17 selected CMIP5 models are included in the multi-model average calculation. In

contrast, there is no inter-model consensus regarding the changes in the impacts of the Indian Ocean Dipole and the Atlantic Niño/Niña on ENSO due to global warming (Fig. S6). This supports the notion that the enhanced impacts of the northwestern hemispheric ENSO precursors under greenhouse warming are not significantly linked to those of the equatorial Indian or Atlantic ENSO precursors.

To understand the physical mechanism of the changes in the multi-model averaged value of the NTA–ENSO and SNP–ENSO relationship, the corresponding changes in the individual models are investigated. This is in line with the assumption that, if the changes in the individual models are constrained to any physical quantity, the same quantity would also constrain the changes in the multi-model averaged value[26,27]. Intriguingly, the degree of the enhancement of the NTA–ENSO relationship is tightly correlated with that of the SNP–ENSO relationship, and vice versa. The scatter plot between the changes in the NTA-regressed ENSO amplitude and those in the SNP-regressed value of an individual model exhibits a strong positive relationship (Fig. 2c). The inter-model correlation between them in the 30 selected CMIP6 models is 0.72 with a p-value of <0.001. These results remain consistent when the 17 selected CMIP5 models are included (i.e., intermodel correlation = 0.65 with a *p* value of <0.001).

## Physical constraints linking the NTA–ENSO and SNP–ENSO relationship

Both NTA and SNP SST anomalies exert remote impacts on the equatorial Pacific SST by expanding the climate signals initially induced over the off-equatorial Pacific. This indicates that the atmospheric response over the off-equatorial Pacific is the key to successfully conveying climate signals led by the northwestern hemispheric ENSO precursors to the equatorial Pacific. In addition to the relationship during any specific period, the changes in the NTA–ENSO or SNP–ENSO relationship are largely determined by the degree of enhancement of the NTA- or SNP-related atmospheric anomalies over the off-equatorial central–eastern Pacific (160° E–120° W, 2° S–8° N) (Fig. 3a, b). The inter-model correlation coefficient between changes in the NTA–ENSO relationship and those in the NTA-regressed precipitation anomalies over the off-equatorial central–eastern Pacific is 0.84 with a *p* value < 0.001 in the 30 selected CMIP6 models. Similarly, for SNP, the correlation is also 0.84 with a *p* value < 0.001.

The wetter climatology over the equatorial[28–30] or the off-equatorial eastern Pacific[31] is known to enhance the in-situ atmospheric responses during ENSO, which eventually enhance the ENSO amplitude[18]. Similarly, since the NTA- and SNP-related atmospheric responses are prominent over the off-equatorial central–eastern Pacific, the wetter mean state over the off-equatorial central–eastern Pacific compared to the other regions plays a crucial role in strengthening the NTA–ENSO and SNP–ENSO connections. To verify this point, the relative wetness over the off-equatorial eastern Pacific is quantified as the difference in the off-equatorial zonal precipitation climatology between the eastern Pacific (150°–90° W, 2° S–8° N) and the western Pacific (130°–150° E, 2° S–8° N) (hereinafter, Off_Zonal_Pr_Diff).

The inter-model relationship between the degree of change in the NTA–ENSO or SNP–ENSO relationship and the Off_Zonal_Pr_Diff is quite robust (Fig. 3c, d); the climate models simulating the wetter mean state over the off-equatorial eastern Pacific in the future climate tend to simulate a stronger enhancement of the NTA–ENSO and SNP–ENSO connections. The inter-model correlation between the changes in the Off_Zonal_Pr_Diff and the intensity change of the NTA–ENSO and SNP–ENSO regressions is 0.49 and 0.58 (Fig. 3c, d), respectively. Both inter-model relationships are significant at the 99% confidence level, suggesting that a single aspect of the mean state change over the off-equatorial eastern Pacific can control the changes in the impact of NTA or SNP SST on ENSO. This conclusion remains valid even after including the 17 selected CMIP5 models.

Since the multi-model averaged response is the sum of the responses of the individual models, it is plausible to conclude that the multi-model averaged change in the NTA–ENSO and SNP–ENSO connections is strongly dependent on the multi-model averaged Off_Zonal_Pr_Diff. This indicates that the enhancement of the NTA–ENSO and SNP–ENSO connections due to global warming is largely caused by the enhancement of the off-equatorial precipitation climatology.

The enhancement of the atmospheric heating anomalies driven by the precipitation-related condensation process over the off-equatorial eastern Pacific leads to cyclonic circulation anomalies[6,31]. As a result, the amplitudes of the equatorial westerlies (Fig. S7) and subtropical southerlies (Fig. S8), which are associated with multiple ENSO precursors tend to intensify with increasing Off_Zonal_Pr_Diff. The enhancement of the NTA- or SNP-related westerly, and southerly anomalies over the subtropical regions of the northern hemisphere enhances the activation of the downwelling Kelvin waves, and the WES feedback, respectively, subsequently increasing ENSO variability (Fig. S9).

The enhanced joint impacts of multiple ENSO precursors have a stronger effect on the frequency of extreme El Niño events compared to the impact of individual ENSO precursors under greenhouse warming. The change in the impacts of NTA and SNP SST on ENSO under greenhouse warming is assessed by dividing the climate models into three groups: models with (1) enhanced NTA–ENSO and SNP–ENSO connections (21 models in CMIP6) (Fig. 4a), (2) weakened NTA–ENSO and enhanced SNP–ENSO connections (9 models in CMIP5/6) (Fig. 4b), and (3) enhanced NTA–ENSO and weakened SNP–ENSO connections (3 models in CMIP5/6) (Fig. 4c). The frequency of extreme El Niño events (defined by E-index > 1.5 s.d., as in ref. 32) increased only in group (1), while the occurrences of extreme El Niño events barely varied for the other groups; in group (1), the number of extreme El Niño events for 50 years significantly increased from 4.1 in the present-day climate to 4.8 in the future climate. This clearly demonstrates that the joint enhancement of the impact of multiple ENSO precursors under greenhouse warming contributes more strongly to the increase in the frequency of extreme El Niño events compared to the enhancement of a single ENSO precursor.

Previous studies have shown that the prediction of ENSO evolution in both dynamical and statistical models can be improved when considering the NTA and SNP SST conditions[7,10,24,33]. Since 1980, all the cases of NTA warming (i.e., 1980, 1988, 1998, and 2010) were followed by a La Niña event[6], and 7 out of the 15 cases of SNP warming (i.e., 1982, 1986, 1991, 1994, 1997, 2014, and 2014) were followed by an El Niño event[34,35]. Given that the impact of NTA and SNP SST on ENSO in enhanced under greenhouse warming, the northwestern hemispheric precursors are expected to contribute to increased forecast skill for ENSO. To examine the changes in ENSO predictability led by the northwestern hemispheric precursors, a statistical model is formulated using multiple linear regression with three predictors. In addition to the NTA and SNP indices, the Niño3.4 index is included as a predictor to account for the self-oscillatory feature of ENSO[36] (see Methods). Note that the training dataset to obtain the regression coefficients is the same as the testing dataset; therefore, this experiment only assesses the potential predictability.

First, the change in the multi-model averaged correlation skill is assessed. The multi-model averaged correlation skill between the predicted Niño3.4 and the reference Niño3.4 (i.e., model simulation) is systematically increases from 0.50 in the present-day climate to 0.58 in the future climate during FMA0 (Fig. 5a). Next, five models with the greatest and weakest changes under Off_Zonal_Pr_Diff are selected. Under greenhouse warming, the ENSO prediction skill of the NTA and SNP SST precursors significantly improves with a nine-month lead for high Off_Zonal_Pr_Diff (Fig. 5b). On the other hand, under low Off_Zonal_Pr_Diff, the prediction skill remains almost the same for present-day and future periods (Fig. 5c). This indicates that

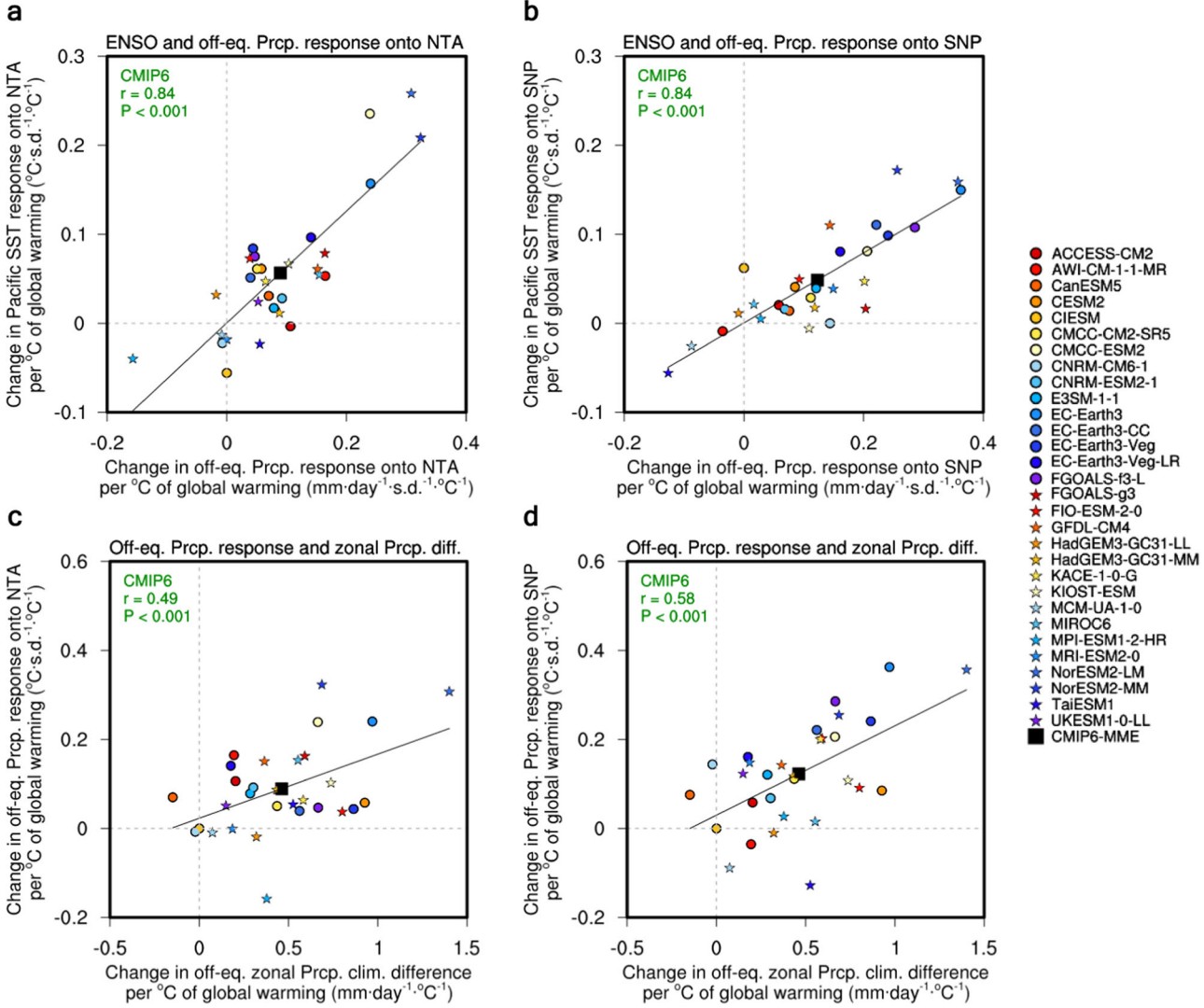

**Fig. 3 | Mechanisms for the stronger northwestern hemispheric El Niño-Southern Oscillation (ENSO) precursors. a, b** Inter-model relationship between changes (future minus present day) of the DOJF1 equatorial Pacific SST response (*y* axis; °C·s.d.⁻¹·°C⁻¹) with the FMAMJJASO0 off-equatorial central–eastern Pacific precipitation (160° E–120° W, 2° S–8° N) response (*x* axis; mm·day⁻¹·s.d.⁻¹·°C⁻¹) for the FMA0 north tropical Atlantic (NTA) SST (**a**) and subtropical northeast Pacific (SNP) SST (**b**). **c, d** Same as **a, b**, but for the FMAMJJASO0 off-equatorial central-eastern Pacific precipitation response (*y* axis; mm·day⁻¹·s.d.⁻¹·°C⁻¹) for FMA0 NTA (**c**) and SNP (**d**) with the FMAMJJASO0 off-equatorial zonal precipitation climatology

difference [(150°–90° W, 2° S–8° N) minus (130°–150° E, 2°S –8° N)] (*x* axis; mm·day⁻¹·°C⁻¹). For the models with available data, the CMIP6 models are denoted by dots and stars (see CMIP6 data in Methods). A linear fit is displayed along with the correlation coefficient *r* and *P* value based on the 30 selected models (black line and green text). The multi-model averaged value for the CMIP6 models (labeled CMIP6-MME) is also denoted by black squares. The changes in each model are scaled by the corresponding increase in the global mean temperature. The regressed coefficients are multiplied by −1 for the NTA SST.

the wetter mean state change over the off-equatorial eastern Pacific and the resulting enhanced role of northwestern hemispheric ENSO precursors in the future climate potentially enhance ENSO forecasting skills.

## Discussion

While previous studies have primarily focused on the changes in the impact of individual ENSO precursors under greenhouse warming, this study highlights the impact of multiple precursors over the northwestern hemisphere on ENSO. The results show that the joint impacts of NTA and SNP on ENSO are significantly enhanced due to global warming and that the change in the impact of multiple precursors on ENSO can be controlled by a single factor: the mean state change over the off-equatorial eastern Pacific. The increase in the precipitation climatology over the off-equatorial eastern Pacific facilitates the induction of in-situ and remote atmospheric responses to the given

NTA and SNP SST anomalies, which eventually contribute to the enhancement of their remote impacts on ENSO. Thus, the enhanced joint impacts of NTA and SNP SST on ENSO under greenhouse warming are largely caused by the change in the wetter mean state over the off-equatorial eastern Pacific.

Our main result is well matched to the recent observed changes; after the 2000s, both the NTA – ENSO and SNP – ENSO connections are intensified with the increased precipitation climatology over the off-equatorial eastern Pacific (Fig. S10). It should be noted that the climatological SST pattern change during the recent observed decades is quite different from that in the model projection results; the zonal SST gradient over the equatorial Pacific is intensified after the 2000s associated with the occurrence of the negative Interdecadal Pacific Oscillations (IPO)[37], while it is decreased in the model projections[38]. This supports our notion that the precipitation is a key indicator to enhance the NTA –, and SNP – ENSO relationship.

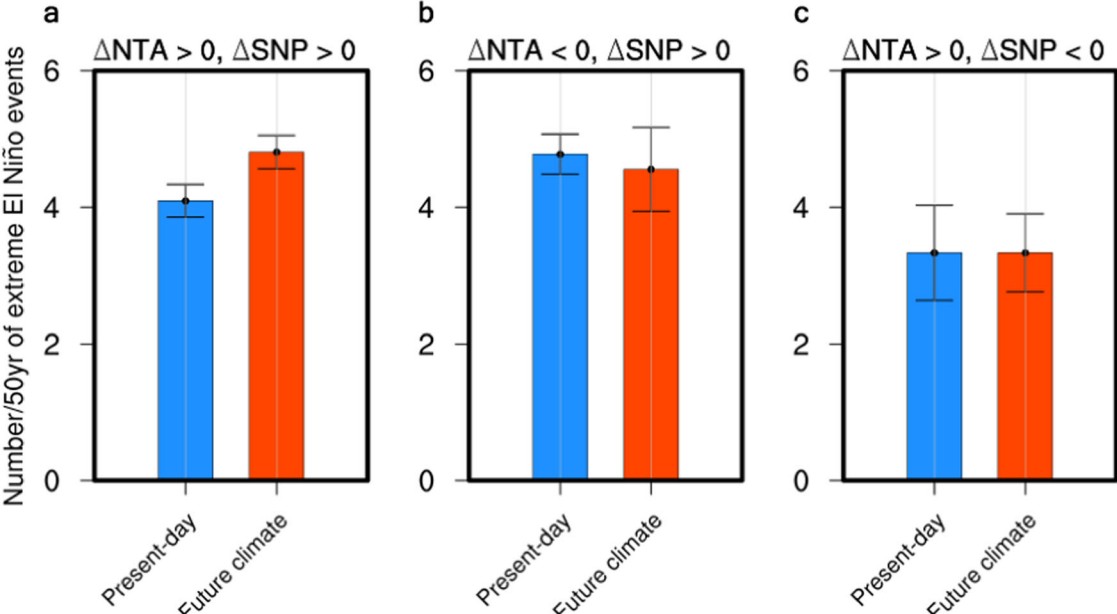

**Fig. 4 | Changes in the occurrences of extreme El Niño events. a** Changes in the number of extreme El Niño events (where the E-index exceeds a 1.5 s.d. value) of the enhanced north tropical Atlantic (NTA)– El Niño-Southern Oscillation (ENSO) ($\triangle$NTA > 0) and subtropical northeast Pacific (SNP)–ENSO ($\triangle$SNP > 0) connections under the present-day climate (1950–1999) and future climate (2050–2099) 50-year periods in the 21 selected CMIP6 models are represented by blue and red bars, respectively; error bars are calculated as 1.0 s.d. of 10,000 inter-realizations using a bootstrap method (see Methods for details on the bootstrap test). **b,c** Same as **a**, but for the weakened NTA–ENSO ($\triangle$NTA < 0) and enhanced SNP–ENSO ($\triangle$SNP > 0) connections in the 9 selected CMIP5/6 models (**b**), and the enhanced NTA–ENSO ($\triangle$NTA > 0) and weakened SNP–ENSO ($\triangle$SNP < 0) connections in the 3 selected CMIP5/6 models (**c**). The frequency of extreme El Niño events under the joint enhancement of the NTA–ENSO and SNP–ENSO connections is projected to increase under greenhouse warming.

The results of this study differ from the weakened impact of NTA SST on ENSO in the future climate observed in CMIP5 models[13,14]. Interestingly, our findings also highlight a systematic difference between the CMIP5 and CMIP6 models to some extent; the CMIP6 models tend to simulate a wetter mean state over the off-equatorial eastern Pacific compared to the CMIP5 models (Fig. S11). This suggests that the varying changes in the impact of NTA SST on ENSO due to global warming between CMIP5 and CMIP6 can be understood within the inter-model framework. Furthermore, it should be noted that the CMIP5 model results do not encompass all possible distributions of simulated mean state changes, but rather represent a subset of the climate model population.

The wetter mean state over the off-equatorial eastern Pacific and the resulting enhanced joint impacts of the northwestern hemispheric ENSO precursors strongly contribute to the increased occurrence of extreme El Niño events under greenhouse warming[39]. Moreover, it is shown that the wetter mean state over the off-equatorial eastern Pacific significantly improves the skill of ENSO prediction by enhancing the impacts of both NTA and SNP SST on ENSO. This signifies that the scientific community is now progressing towards understanding the impact of multiple precursors on ENSO, with the ultimate goal of devising a comprehensive strategy for climate change adaptation. This strategy aims to considers not only the overall variability and predictability of ENSO but also its extreme occurrences.

## Methods
### Observed and CMIP5/6 data
To characterize the NTA–ENSO and SNP–ENSO connections, the monthly mean SST data from the Extended Reconstructed Sea Surface Temperature version 5 (ERSST v5) dataset[40] were used. The monthly mean precipitation and low-level winds were obtained from the European Centre for Medium-Range Weather Forecasts' fifth generation of European Reanalysis (ECWMF–ERA5) dataset[41]. Anomalies were calculated by subtracting the monthly mean climatology over the

observed period (1950–2020). To eliminate any potential influences associated with global warming, a linear trend was removed.

This study utilized monthly outputs from 17 selected CMIP5 models forced by historical forcing up to 2005 and the subsequent representative concentration pathway (RCP) 8.5 scenario (an escalating radiative force throughout the twenty-first century, reaching approximately 8.5 W m$^{-2}$ in 2100)[42], covering the period from 1950 to 2100. Additionally, 30 selected CMIP6 models were used forced by historical forcing up to 2014 and the subsequent Shared Socioeconomic Pathway 5–8.5 scenario (approximately equivalent to RCP 8.5)[43]. One ensemble member of each model was used, specifically r1i1p1 in CMIP5 and mostly r1i1p1f1 in CMIP6 (Table S1). The NTA and SNP SST forcings on ENSO between the present-day and future climates were compared. All the observational and modeling data were interpolated to a 2.5° × 2.5° grid. The anomalies were computed by removing seasonal cycles and quadratic trends from the monthly data.

The anomalies in the CMIP5 and CMIP6 outputs were obtained based on the full 150-year period (1950–2099). No evidence was found to suggest that the results from the 47 selected CMIP5 and CMIP6 models are dependent samples; due to butterfly effects[32], even results from large ensemble experiments with the same model can be treated as independent realizations.

### Depiction of the NTA and SNP index
In observations and the CMIP5/CMIP6 models, the NTA and SNP indices during FMA0 were defined by taking the area-averaged SST anomalies that are quadratically detrended over NTA (80° W–20° E, 0°–15° N)[6] and SNP (170°–120° W, 5°–25° N)[7], respectively. Following previous studies[6,8], the ENSO signal was removed using linear regression with respect to the CTI index (SSTs averaged over 180°–90° W, 6° S–6° N) during the preceding winter (D − 1JF0, where "−1" denotes the preceding year) for the FMA0 NTA index and the simultaneous early spring (FMA0) for the FMA0 SNP index.

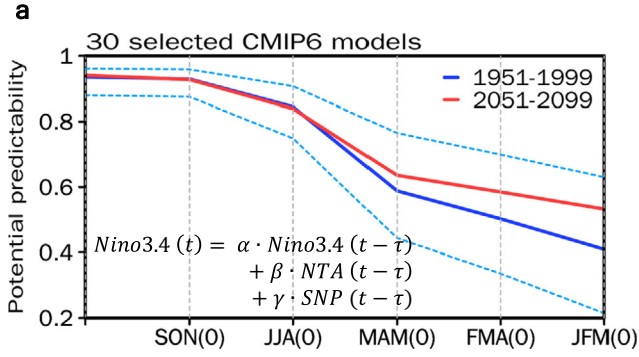

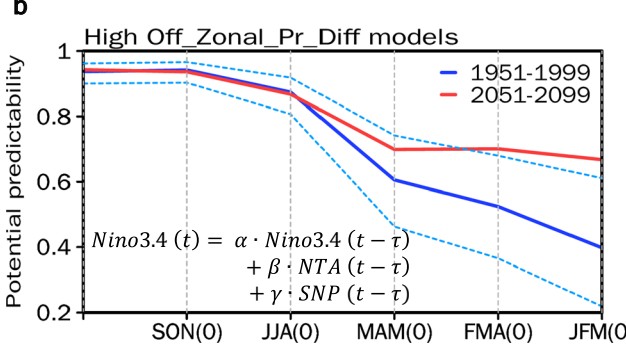
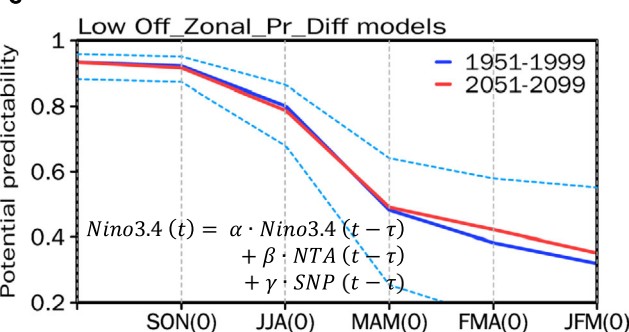

**Fig. 5 | Potential predictability of El Niño-Southern Oscillation (ENSO) using the northwestern hemispheric precursors. a** Correlation between the observed DOJF1 Niño3.4 SST from multiple regression with preceding predictors at a time of $(t-\tau)$, where $\tau$ represents a different lead time for each predictor, as identified by the regression coefficients, under the present-day climate (1951–1999; blue line) and future climate (2051–2099; red line) in the 30 selected CMIP6 models. Dashed lines indicate the multi-model s.d. of the 30 selected CMIP6 models under the

present-day climate. **b,c** Same as **a**, but for the 5 selected CMIP6 models by the greatest (CESM2, EC-Earth3, EC-Earth3-Veg, FIO-ESM-2-0, and NorESM2-LM) (**b**) and weakest (CanESM5, CNRM-CM6-1, CIESM, MCM-UA-1-0, and UKESM1-0-LL) (**c**) changes in the FMAMJJASO0 Off_Zonal_Pr_Diff. In addition to their own values, these predictors include the Niño3.4, the north tropical Atlantic (NTA), and the subtropical northeast Pacific (SNP) values, with regression coefficients $\alpha$, $\beta$, and $\gamma$ determining their relative contributions using multiple regression, respectively.

## Bootstrap and significance testing

The Bootstrap test was conducted to examine the significance of the ensemble mean change[44]. A total of 10,000 realizations were performed to calculate the mean from the 30 (17) selected CMIP6 (CMIP5) models. Each realization involved averaging over 30 (17) samples that were independently and randomly resampled from the 30 (17) selected models. During this random resampling process, any model could be selected multiple times. The standard deviation (SD) of the 10,000 realizations was calculated for each period. If the difference in mean values between the future and present-day periods exceeded the sum of the two separate SD values from the 10,000 realizations, then the change was statistically significant at the 95% confidence level. The two-tailed Student's $t$-test for the statistical significance tests[45] was performed for the regression and correlation analyses.

## Partial regression

To obtain the "pure" influence of NTA and SNP, a partial regression analysis was conducted to exclude the influence of ENSO. The partial regression coefficient can be described as follows:

$$C_{Y(B|A)} = \frac{R_{YB} - R_{YA} * R_{AB}}{\sqrt{1 - R_{AB}^2}} * \frac{S_Y}{S_{B|A}} \quad (1)$$

where $C_{Y(B|A)}$ represents the partial regression coefficient between the target variable $Y$ and the NTA and SNP index ($B$) after the removal of the influence of ENSO ($A$). $S_Y$ and $S_{B|A}$ represent the SD of variable $Y$ and the NTA and SNP index ($B$) after the removal of the CTI index ($A$), respectively. The degree of the impact of NTA or SNP SST is quantified by the partial lagged regression coefficient of the quadratically

detrended equatorial Pacific SST anomalies (140° E–80° W, 5° S–5° N) during DOJF1 onto the FMA0 NTA and SNP SST indices after removing the linearly regressed ENSO signal during the preceding winter (D − 1JF0) and simultaneous early spring (FMA0).

## Partial nudging experiments using a Coupled Global Climate Model (CGCM)

To investigate the impacts of the NTA and SNP SST on the subsequent ENSO, the Community Earth System Model, version 1 (CESM1)[46] was used for the idealized CGCM experiments. The model resolution was a 1.9° × 2.5° grid with the standard 30 vertical levels for the atmosphere and an approximate 1° grid for the ocean.

We conducted two sets of partial nudging experiments. The nudging scheme is one of the simplest data assimilation techniques and can be described as follows:

$$\frac{dSST}{dt} = A + \frac{(SST_{obs} - SST)}{\tau} \quad (2)$$

where $SST$, and $A$ represent the $SST$ in the CGCM, and the conventional dynamical and physical terms to lead SST tendency, respectively. The second term on the right-hand side is the nudging term. By adding the nudging term, the simulated SST always moves toward the observed SST. The nudging time scale ($\tau$) was prescribed as 1 day so that the simulated SST state in the NTA and SNP was strongly nudged toward the given observed SST state.

In the first experiment, both negative NTA and positive SNP SST anomalies were nudged in the north tropical Atlantic (80° W − 20°E, 0° − 15° N) and the subtropical North Pacific (170° − 120°W, 5° − 25° N)

with the seasonally varying climatological SSTs during the spring season (February-March-April-May (FMAM0)), while the ocean-atmosphere was freely coupled outside of the prescribed regions (Exp_NTA + SNP). The negative NTA- and positive SNP-related SST anomalies were obtained by regressing the observed SST anomalies during the FMAM with respect to simultaneous negative NTA and positive SNP index using ERSST v5 for 1951–2010, respectively. As a reference, an identical experiment was performed with seasonally varying climatological SSTs (Exp_CTRL). The differences between the Exp_NTA + SNP and Exp_CTRL can be considered as the impacts of the NTA and SNP SST on the subsequent ENSO, irrelevant to previous ENSO conditions. Both experiments were forced by a preindustrial atmospheric composition in 1850 (constant $CO_2$ concentration of 284 ppm). A total of 30 ensemble members with different initial conditions were employed for both experiments. Each experiment was run for 11 months from February to December.

## Data availability
All datasets related to this paper are publicly available. The ERSSTv5 can be accessed at https://psl.noaa.gov/data/gridded/data.noaa.ersst.v5.html. The ECMWF – ERA5 dataset are available at https://cds.climate.copernicus.eu/#!/search?text=ERA5&type=dataset. The CMIP5 dataset provided by ESGF can be obtained from the open-source link: https://esgf-index1.ceda.ac.uk/search/cmip5-ceda/. To download the required variables, select Frequency as monthly, Experiment ID as *historical or rcp85*, Ensemble as *r1i1p1*, Model as CMIP5 models used for analyses, and then download the nc files by opening them one by one from the list that appears as search outputs. For the CMIP6 datasets, the open-source link is https://esgf-node.llnl.gov/search/cmip6/. The process is similar to obtaining CMIP5, but select Experiment ID as *historical or ssp585* and Variant Label as mostly *r1i1p1f1*. Detailed references and DOI URLs for each CMIP6 model can be found in the Table S1.

## Code availability
The data used in this study were analyzed using the NCAR Command Language (NCL; http://www.ncl.ucar.edu/). The code of the CESM1.2 model used in this study is available at http://www.cesm.ucar.edu/models/cesm1.2. The codes of the Bootstrap method can be accessed at https://www.ncl.ucar.edu/Applications/bootstrap.shtml, and the codes for the Student's *t*-test for the significance testing can be found at https://doi.org/10.5281/zenodo.8348132. All relevant codes used in this study are available, upon request, from the corresponding author.

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

## Acknowledgements
This research was supported by the Basic Science Research Program through the National Research Foundation of Korea (NRF) funded by the Ministry of Education (NRF-2021R1I1A1A01053579). Y. -G. Ham is supported by the Korea Meteorological Administration Research and Development Program under Grant KMI2021-01513, and the National Research Foundation of Korea (NRF)(NRF-2020R1A2C21010025).

## Author contributions
H-S.J. and Y-G.H. designed the research; H-S.J. conducted the data analysis, prepared the figures, performed the experiments, and wrote the initial manuscript. H-S.J. and Y-G.H. made significant contributions to the interpretation of the analysis and actively participated in the discussions of the results and reviewed the manuscript.

## Competing interests
The authors declare no competing interests.
