## [Peer Review File · Nature Communications]

REVIEWER COMMENTS

Reviewer #1 (Remarks to the Author):

Review of „Enhanced joint impact of the western hemispheric precursors on the El Niño–Southern Oscillation under greenhouse warming“ by Hyun-Su Jo and Yoo-Geun Ham

This manuscript looks at statistical relationships between sea surface temperature anomalies (SSTA) in the northern subtropical Pacific (SNP) and Northern subtropical Atlantic (TNA) on El Niño–Southern Oscillation (ENSO) in coupled climate models (CMIP5 and CMIP6 collections). The authors find statistically significant relationships between boreal spring SSTA in the two considered regions and ENSO in the subsequent boreal winter, and they find a strengthening of the found relationships with projected warming of the climate. They furthermore postulate that those models that project a strengthening relationship between both SNA and TNA SSTA with ENSO also project an increase of extreme El Niño events in a warmer future.

While the manuscript is well-written my main difficulty is that it mixes causality and correlation. The latter is not sufficient to prove the former. The found lagged regressions are indeed strong and appear statistically robust, but it does not prove the direction of the interaction (SNP and TNA force ENSO or vice versa?). The fact that the relationships are found when ENSO is lagging the predictors is not sufficient as a proof, as a recent paper (Zhang et al. 2021, which is not cited in the present manuscript) has nicely demonstrated (both statistically and with model experiments) that the apparent lagged response of ENSO to TNA SSTA is merely a result of the autocorrelation of ENSO itself (an El Niño event tends to be succeeded by a La Niña event the following boreal winter).

I understand that the authors of the present manuscript employ a partial lagged regression method that attempts to statistically remove the influence of ENSO itself on the diagnosed relationship between TNA/SNP SSTA on ENSO. However, this method can only remove the first-order linear relationship between ENSO and TNA/SNP, but cannot remove non-linearities, e.g., arising from asymmetric responses to El Niño or La Niña events. Another factor hampering the partial regression method is that it is not able to account for temporally varying spectral characteristics of ENSO within a single model (e.g., decadal changes from biennial to lower-frequency oscillations) that will also change the lead-lag-relationships of ENSO with SNP and TNA SSTA (in the sense of the impact of ENSO on those regions). It thus cannot be ruled out that the found relationships arise from residuals of the ENSO-to-SNP/TNA connection that the statistical method is not able to fully remove.

Hence, I conclude from my assessment of the present manuscript that the postulated relationships (including the postulated directionality, namely SNP/TNA SSTA forcing ENSO and not the other way round) and their changes may well be true, but the employed statistical method (partial regression) is not sufficient to rule out the possibility that the found relationships are spurious (as suggested by Zhang et al. 2021). In my opinion tailored model experiments that allow for a clear demonstration of the TNA/SNP-to-ENSO impact would be needed to corroborate the statistical findings presented in this manuscript. Given that (in my opinion) the manuscript would require inclusion of substantial additional material to become convincing, I refrain from including specific comments on it in its present form.

Reference

Zhang, W., Jiang, F., Stuecker, M.F. et al. Spurious North Tropical Atlantic precursors to El Niño. *Nat Commun* 12, 3096 (2021). <https://doi.org/10.1038/s41467-021-23411-6>

Reviewer #2 (Remarks to the Author):

This study investigates the impacts of the NTA and SNP (NPMM) regions as precursors of ENSO. Using CMIP5 and CMIP6 models, the authors find that the impacts of the NTA and SNP are enhanced in the future and that this is closely linked to a projected El-Nino like mean-state precipitation response. The study is well presented and in my opinion the methodology is sound. This work provides useful insights into the dynamics behind projected increases in extreme El Nino events. However there are some potential caveats, in particular regarding known model biases, that the authors should consider addressing.

Firstly, in several CMIP models, there is quite a large bias in the seasonal cycle of ENSO and in the Atlantic Meridional Mode's relationship with tropical Atlantic SST/wind (e.g. Mckenna et al 2020, Xia et al 2023). Given that your results are quite seasonally dependent, have you checked that the models in your subset are not impacted by these biases?

Secondly, it would be good for the authors to comment on (or at least note as a caveat) that the El Nino-like warming pattern projected by most CMIP models is seemingly at odds with the observed La Nina-like pattern in recent decades (e.g. Lee et al 2022, Seager et al 2019).

Minor comments:

L30: differ -> differs

L57: unclear what 'intermediate climate model' is referring to.

L93: 'Totally' -> 'In total'

L223: 'improves' -> 'is enhanced'

References

McKenna, S., Santoso, A., Gupta, A.S. et al. Indian Ocean Dipole in CMIP5 and CMIP6: characteristics, biases, and links to ENSO. *Sci Rep* 10, 11500 (2020). <https://doi.org/10.1038/s41598-020-68268-9>

Xia F, Zuo J, Sun C, Liu A. The Atlantic Meridional Mode and Associated Wind–SST Relationship in the CMIP6 Models. *Atmosphere*. 2023; 14(2):359. <https://doi.org/10.3390/atmos14020359>

Seager, R., Cane, M., Henderson, N. et al. Strengthening tropical Pacific zonal sea surface temperature gradient consistent with rising greenhouse gases. *Nat. Clim. Chang.* 9, 517–522 (2019). <https://doi.org/10.1038/s41558-019-0505-x>

Lee, S., L’Heureux, M., Wittenberg, A.T. et al. On the future zonal contrasts of equatorial Pacific climate: Perspectives from Observations, Simulations, and Theories. *npj Clim Atmos Sci* 5, 82 (2022). <https://doi.org/10.1038/s41612-022-00301-2>

REVIEWER COMMENTS:

Reviewer #1:

This manuscript looks at statistical relationships between sea surface temperature anomalies (SSTA) in the northern subtropical Pacific (SNP) and Northern subtropical Atlantic (TNA) on El Niño-Southern Oscillation (ENSO) in coupled climate models (CMIP5 and CMIP6 collections). The authors find statistically significant relationships between boreal spring SSTA in the two considered regions and ENSO in the subsequent boreal winter, and they find a strengthening of the found relationships with projected warming of the climate. They furthermore postulate that those models that project a strengthening relationship between both SNA and TNA SSTA with ENSO also project an increase of extreme El Niño events in a warmer future.

While the manuscript is well-written my main difficulty is that it mixes causality and correlation. The latter is not sufficient to prove the former. The found lagged regressions are indeed strong and appear statistically robust, but it does not prove the direction of the interaction (SNP and TNA force ENSO or vice versa?). The fact that the relationships are found when ENSO is lagging the predictors is not sufficient as a proof, as a recent paper (Zhang et al. 2021, which is not cited in the present manuscript) has nicely demonstrated (both statistically and with model experiments) that the apparent lagged response of ENSO to TNA SSTA is merely a result of the autocorrelation of ENSO itself (an El Niño event tends to be succeeded by a La Niña event the following boreal winter).

[Reply] Thank you for the careful reading and valuable comments. We have revised the manuscript to fully address the reviewer's concerns. In order to provide further insights, we conducted additional idealized model experiments using the Community Earth System Model version 1 (CESM1). The results of these experiments confirm that the impacts of both NTA and SNP on ENSO significantly increase under greenhouse warming.

Major comments

1. I understand that the authors of the present manuscript employ a partial lagged regression method that attempts to statistically remove the influence of ENSO itself on the diagnosed relationship between TNA/SNP SSTA on ENSO. However, this method can only remove the first-order linear relationship between ENSO and TNA/SNP, but cannot remove non-linearities, e.g., arising from asymmetric responses to El Niño or La Niña events. Another factor hampering the partial regression method is that it is not able to account for temporally varying spectral characteristics of ENSO within a single model (e.g., decadal changes from biennial to lower-frequency oscillations) that will also change the lead-lag-relationships of ENSO with SNP and TNA SSTA (in the sense of the impact of ENSO on those regions). It thus cannot be ruled out that the found relationships arise from residuals of the ENSO-to-SNP/TNA connection that the statistical method is not able to fully remove.

Hence, I conclude from my assessment of the present manuscript that the postulated relationships (including the postulated directionality, namely SNP/TNA SSTA forcing ENSO

and not the other way round) and their changes may well be true, but the employed statistical method (partial regression) is not sufficient to rule out the possibility that the found relationships are spurious (as suggested by Zhang et al. 2021). In my opinion tailored model experiments that allow for a clear demonstration of the TNA/SNP-to-ENSO impact would be needed to corroborate the statistical findings presented in this manuscript. Given that (in my opinion) the manuscript would require inclusion of substantial additional material to become convincing, I refrain from including specific comments on it in its present form.

[Reply] We understand the reviewer's concerns regarding the spurious identification of northwestern hemispheric precursors for ENSO due to the oscillatory feature of ENSO. To address these issues, we conducted two sets of partially-coupled experiments using the CESM1. First, we performed an experiment referred to as the Exp_NTA+SNP, which both negative NTA and positive SNP SST anomalies were nudged in the north tropical Atlantic (80° W– 20° E, 0° – 15° N) and the subtropical North Pacific (170° – 120° W, 5° – 25° N) with seasonal varying climatological SSTs during the spring season (February-March-April-May (FMAM0)), while the ocean-atmosphere was freely coupled outside of the prescribed regions. The negative NTA- and positive SNP-related SST anomalies were obtained by regressing the observed SST anomalies during the FMAM with respect to simultaneous negative NTA (SST anomalies averaged over 80° W– 20° E, 0° – 15° N) and positive SNP index (SST anomalies averaged over 170° – 120° W, 5° – 25° N) using ERSST v5 for 1951–2010, respectively. As a reference, an identical experiment was performed with seasonally varying climatological SSTs (Exp_CTRL). The differences between the Exp_NTA+SNP and Exp_CTRL can be considered as the impacts of the NTA and SNP SST on the subsequent ENSO, irrelevant to previous ENSO conditions. Both experiments were forced by a pre-industrial atmospheric composition in 1850 (constant CO₂ concentration of 284 ppm). A total of 30 ensemble members with different initial conditions were employed for both experiments. Each experiment was run for 11 months from February to December. Nudging time scale was prescribed as one day.

Figure 1-A displays the difference between the Exp_NTA+SNP and Exp_CTRL. The simulation clearly demonstrates that the negative NTA and positive SNP SST anomaly forcing can lead to a significant El Niño-like signal over the equatorial Pacific in the subsequent winter. Equatorial westerly anomaly induced by the NTA and SNP SST over the western-central Pacific during the FMA and JJA seasons contributes to induce the warm SST anomalies over the equatorial central-eastern Pacific during the OND season. The simulation results exhibit striking similarities to the observed patterns (Figure 1-B), indicating that the model realistically captures the impact of NTA and SNP SST on the evolution of ENSO.

It should be emphasized that this approach (i.e., multi-member-based partial nudging

experiments) has been widely employed using various CGCMs to demonstrate that the NTA or SNP SST anomalies can lead to the ENSO in subsequent seasons (see Figure 1 of Jia et al., 2016 and Figure 5 of Jia et al., 2021) (Ham et al., 2013; Jia et al., 2016; Jia et al., 2021; Jiang and Li, 2021; Jiang et al., 2022; Ding et al., 2023). This alleviates the concerns that might come from the single-model-based partial nudging experiments in this rebuttal letter.

The aforementioned discussions are added in the revised manuscript as follows, and the Figure 1-A is added as Supplementary Figure 5.

Line 127-130: “Those impacts of both NTA and SNP on the subsequent ENSO are confirmed by a set of partially-coupled model experiments by imposing NTA and SNP SST anomaly forcing (see Methods for details) (Fig. S5), which would reject the possibility that the NTA-, and SNP-related ENSO is a statistical artifact due to the oscillatory feature of the ENSO (Zhang et al., 2021).”

Figure 1-A. Idealized partially-coupled model experiments. Changes in the SST (shading, °C) and 925hPa wind (vector, $\text{m}\cdot\text{s}^{-1}$) anomalies in the north tropical Atlantic (NTA) and subtropical North Pacific (SNP) experiment (Exp_NTA+SNP) compared to the control experiment (Exp_CTRL) under a pre-industrial CO_2 concentration (284 ppm) during the (a) FMA, (b) JJA and (c) OND seasons. The shading and vector denote regions where the statistical significance is above the 90% confidence level.

Figure 1-B. Observed time evolution of the NTA- and SNP-induced SST anomalies. (a-c) Lag regression of SST ($^{\circ}\text{C}\cdot\text{s.d.}^{-1}$; contours and shading) anomalies onto the FMA0 SNP SST during the FMA0, MJJ0, and ASO0 seasons based on observations. (d-f) Same as (a-c) but for the NTA-induced SST anomalies. The shading represents regression coefficients that are statistically significant at the 95% confidence level. The regressed coefficients are multiplied by -1 for the NTA SST.

To further demonstrate that the response of the NTA and SNP SST anomalies on the ENSO is irrelevant to the preceding Warm Water Volume (WWV) signal, Figure 1-C displays the histogram of the difference of the Niño3.4 index during the OND season for 30 ensemble members with respect to the difference in the tropical ocean heat content in February between the Exp_NTA+SNP and Exp_CTRL. The histogram shows that the amplitude of the El Niño events led by the NTA and SNP SST anomalies is irrelevant to the preceding WWV. For example, the El Niño response to the given NTA and SNP SST anomalies exhibits a strongest amplitude for both the positive and negative extreme bin of the WWV. This suggests that the simulation of the El Niño event as a response to the NTA and SNP SST anomalies is not favored by any specific phase of the preceding WWV signal.

Figure 1-C. Histogram of 30 ensemble members of the difference between the Exp_NTA+SNP and Exp_CTRL for the OND0 ENSO response to the Feb0 Pacific Ocean Heat Content (OHC). The ENSO ($^{\circ}\text{C}$) response is measured by the OND0 Niño3.4 index (170°W – 120°W , 5°S – 5°N), and the OHC ($\times 10^8 \text{ J}\cdot\text{m}^{-2}$) is averaged over the equatorial Pacific (120°E – 80°W , 5°S – 5°N) during the Feb0.

References

- Ding, R., Nnamchi, H. C., Yu, J.-Y., Li, T., Sun, C., Li, J., et al. (2023). North Atlantic oscillation controls multidecadal changes in the North Tropical Atlantic– Pacific connection. *Nat. Commun.*, *14*(1), 1-10.
- Ham, Y.-G., Kug, J.-S., Park, J.-Y., & Jin, F.-F. (2013). Sea surface temperature in the north tropical Atlantic as a trigger for El Niño/Southern Oscillation events. *Nature Geosci.*, *6*(2), 112-116.
- Jia, F., Cai, W., Gan, B., Wu, L., & Di Lorenzo, E. (2021). Enhanced North Pacific impact on El Niño/Southern Oscillation under greenhouse warming. *Nature Clim. Change*, *11*(10), 840-847.
- Jia, F., Wu, L., Gan, B., & Cai, W. (2016). Global warming attenuates the tropical Atlantic-Pacific teleconnection. *Scientific reports*, *6*(1), 1-7.
- Jiang, L., & Li, T. (2021). Impacts of tropical North Atlantic and equatorial Atlantic SST Anomalies on ENSO. *J. Clim.*, *34*(14), 5635-5655.
- Jiang, L., Li, T., & Ham, Y. G. (2022). Critical role of tropical North Atlantic SSTA in boreal summer in affecting subsequent ENSO evolution. *Geophys. Res. Lett.*, e2021GL097606.

REVIEWER COMMENTS:

Reviewer #2:

This study investigates the impacts of the NTA and SNP (NPMM) regions as precursors of ENSO. Using CMIP5 and CMIP6 models, the authors find that the impacts of the NTA and SNP are enhanced in the future and that this is closely linked to a projected El-Nino like mean-state precipitation response. The study is well presented and in my opinion the methodology is sound. This work provides useful insights into the dynamics behind projected increases in extreme El Nino events. However, there are some potential caveats, in particular regarding known model biases, that the authors should consider addressing.

[Reply] We appreciate the reviewer for their careful reading and invaluable comments. To address the reviewer's concern regarding known model biases, we have conducted additional analyses and revised to certain sections of the manuscript. The revised manuscript also incorporates new sentences and figures to specifically address the requests made by the reviewer.

Major comments

1. Firstly, in several CMIP models, there is quite a large bias in the seasonal cycle of ENSO and in the Atlantic Meridional Mode's relationship with tropical Atlantic SST/wind (e.g. Mckenna et al 2020, Xia et al 2023). Given that your results are quite seasonally dependent, have you checked that the models in your subset are not impacted by these biases?

[Reply] Thank you for the reviewer's comment. To address the reviewer's comments, we firstly examined a simulation quality of the seasonal cycle of ENSO, NTA, and SNP by comparing the simulated and observed standard deviations of the Niño3.4, NTA, and SNP indices for each calendar month (Figure 2-A). In terms of ENSO phase-locking, the multi-model ensemble (MME) values exhibited a similar behavior to the observed; it showed a peak variability in December and January, even though the seasonal STD difference is slightly weaker than the observed (Figure 2-A(a) and 2-A(d)). This suggests that the simulation quality of ENSO phase-locking is reasonably good in both CMIP5 and CMIP6, which is consistent with a previous study (Ham and Kug, 2014).

Regarding the NTA, the amplitude of the observed NTA seasonal cycle is to a large extent comparable to those of the CMIP5 and CMIP6 MME with its peak during boreal spring (Figure 2-B(b) and 2-B(e)). The CMIP6 MME performs well in simulating the seasonality of the NTA with the peak amplitude appearing during boreal spring as in the observations, while the variation simulated in the CMIP5 MME show nearly flat month-to-month variation without any obvious peak, suggesting that the majority of CMIP5 models fail to simulate the seasonality

and peak of the observed NTA. This suggests that, compared with the CMIP5 models, the CMIP6 models have substantially improved in simulating the seasonality and peak of the NTA.

As for the SNP, the greatest variability occurs during boreal spring in both observations and the CMIP5 and CMIP6 models (Figure 2-B(c) and 2-B(f)). This implies that the observed SNP phase-locking is generally well-represented by the CMIP5/6 models with a reasonable model spread; the seasonal variations of the MME values most overlap that of the observations with the peaks during boreal spring.

To quantitatively evaluate a model's performance in terms of ENSO, NTA, and SNP phase-locking, we defined a phase-locking performance index (referred to as the PP index, Ham and Kug, (2014)), as the correlation coefficient between the standard deviations of the Niño3.4, NTA, and SNP indices for each model and observation in each calendar month. Figure 2-B presents the PP index of CMIP5 and CMIP6 models for the Niño3.4, NTA, and SNP indices, showing predominantly positive values across the majority of CMIP models, except for a few models. This supports our previous notion that most CMIP models tend to accurately simulate the ENSO, NTA, and SNP phase-locking, as in the observations. We selected the CMIP models whose three PP indices (i.e., PP index for ENSO, NTA, and SNP index) exceed 0.3, then, re-evaluated the impacts of NTA and SNP SST on ENSO under greenhouse warming with the selected models (Figure 2-C). The multi-model averaged value in the future climate for the 21 selected models still shows an increase compared to that in the present-day climate (Figure 2-C(a) and 2-C(b)). Additionally, a strong positive correlation between the degree of enhancement of the NTA–ENSO relationship and that of the SNP–ENSO relationship is still rigorous (Figure 2-C(c)), supporting that our results are not influenced by model biases. We included Figure 2-A and 2-B as Supplementary Figure 1 and Figure 2, and the related descriptions are added to the revised manuscript as follows:

Line 102-105: “It should be noted that the selected CMIP5/6 models accurately simulate the phase-locking of ENSO, NTA, and SNP as the observed to some extent (Figs. S1 and S2), implying that the physical mechanism deriving those climate variabilities would be realistic.”

Figure 2-A. Monthly standard deviation (SD) of the (a) Niño3.4, (b) NTA, and (c) SNP indices in the 17 selected CMIP5 models. (d-f) Same as (a-c) but for the 30 selected CMIP6 models. The *black curves* represent the multi-model ensemble (MME) values, and the *green curves* represent the observations. To focus on the seasonal variation, the values are normalized by the total respective SD of the indices using all months.

Figure 2-B. Phase-locking Performance (PP) index of the 17 selected CMIP5 models for the (a) Niño3.4, (b) NTA, and (c) SNP indices, which is defined as the correlation coefficient between the observed and simulated monthly standard deviations of Niño3.4, NTA, and SNP, respectively. (d-f) Same as (a-c) but for the selected 30 CMIP6 models. The *red bars* represent correlation coefficients above 0.3.

Figure 2-C. (a) Comparison of the D0JF1 equatorial Pacific SST response ($^{\circ}\text{C}\cdot\text{s.d.}^{-1}$) to the FMA0 NTA SST under the present-day (1951–1999; blue bars) and future climate (2051–2099; red bars) in the 16 selected CMIP6 models. The multi-model averaged value and corresponding error bars for the 16 selected CMIP6 models (labeled CMIP6-MME) and the combined 21 selected CMIP5/6 models combined (labeled CMIP5&6-MME) are also shown. (b) Same as a but for the FMA0 SNP SST. Error bars are calculated as 1.0 s.d. of 10,000 inter-realizations of a bootstrap method (see Methods for details on the bootstrap test). (c) Inter-model relationship between changes (future minus present day) of the D0JF1 equatorial Pacific SST response to the respective FMA0 SNP SST (y axis; $^{\circ}\text{C}\cdot\text{s.d.}^{-1}\cdot^{\circ}\text{C}^{-1}$) and FMA0 NTA SST (x axis; $^{\circ}\text{C}\cdot\text{s.d.}^{-1}\cdot^{\circ}\text{C}^{-1}$). For the models with available data, the CMIP5 models are denoted by triangles and crosses and the CMIP6 models are denoted by dots and stars (see CMIP5/6 data in Methods). A linear fit is displayed along with the correlation coefficient r and P value based on the 16 selected CMIP6 models (green text) and the 21 selected CMIP5/6 models (black line and black text). The multi-model averaged value for the CMIP5/6 models (labeled CMIP5&6-MME) is also denoted by black squares. The changes in each model are scaled by the corresponding increase in the global mean temperature. The regressed coefficients are multiplied by -1 for the NTA SST.

2. Secondly, it would be good for the authors to comment on (or at least note as a caveat) that the El Niño-like warming pattern projected by most CMIP models is seemingly at odds with the observed La Niña-like pattern in recent decades (e.g. Lee et al 2022, Seager et al 2019).

[Reply] We greatly appreciate the reviewer's comment. As the reviewer pointed out, most climate models predict a weakening of the west-to-east warm-to-cool sea surface temperature (SST) gradient across the equatorial Pacific (El Niño-like mean state change), whereas the observed SST gradient is strengthened in recent decades, associated with the occurrence of the negative Interdecadal Pacific Oscillations (IPO). While the long-term SST change exhibited the distinct patterns between the observations and global warming simulations, the mean precipitation over the off-equatorial Pacific, which we emphasized as a key factor determining the changes in the strength of the NTA- or SNP-related atmospheric teleconnections, showed a consistent increase in time; the area-averaged mean precipitation over the off-equatorial eastern Pacific (150° – 90° W, 2° S– 8° N) during the spring-to-fall seasons (February to October) reveals an increase in both recent observed periods and future periods in CMIP6 models (Figure 2-D).

The observed increases in the off-equatorial mean precipitation can be understood by examining the detailed spatial distribution of the negative IPO-related SST anomalies; while the negative SST anomalies are overwhelmed over the tropical eastern Pacific during the negative IPO, its amplitude tends to exhibit a local minimum over the off-equatorial Pacific (see Figure 2 and Figure 3 of Medhaug et al., 2017). Likewise, the observed mean SST changes exhibited weaker negative values over the off-equatorial Pacific than those over the adjacent regions in recent decades (Figure 2-E(a)). This 'relative' warming over the off-equatorial eastern Pacific during the negative IPO can lead to the upward motion to compensate the downward motions over the overall eastern Pacific due to the stronger negative SST anomalies, which eventually increase the precipitation (Figure 2-E(b)).

Then, how did the NTA- or SNP-related ENSO amplitude change in the observations? Figure 2-E displays the 21-yr running correlation coefficients of the D0JF1 Niño3.4 index with the preceding FMA0 NTA and SNP indices. The results indicate that the relationship between NTA–ENSO and SNP–ENSO has become stronger over the past decades, consistent with previous studies (Wang et al., 2017; Fan et al., 2022). This demonstrates that the relationship between the climatological precipitation changes and the NTA-, and SNP-ENSO relationship in the model simulations are consistently shown in the observations.

As the key climatological factor, which is emphasized in this study, is the precipitation,

rather the SST, and the statement about the SST changes can confuse the readers in mingling our results to the observed changes, we realize that the term “El Niño-like mean state” is not appropriate for conveying our main finding. That is, to avoid confusion, we have removed the term “El Niño-like mean state” and replaced it with “the wetter mean state over the off-equatorial eastern Pacific” throughout the revised manuscript. These points are emphasized in the revised manuscript as follow, and combined Figure 2-D and 2-F is added as Supplementary Figure 10.

Line 262-269: “Our main result is well matched to the recent observed changes; after the 2000s, both the NTA–ENSO and SNP–ENSO connections are intensified with the increased precipitation climatology over the off-equatorial eastern Pacific (Fig. S10). It should be noted that the climatological SST pattern change during the recent observed decades is quite different from that in the model projection results; the zonal SST gradient over the equatorial Pacific is intensified after the 2000s associated with the occurrence of the negative Interdecadal Pacific Oscillations (IPO) (Meehl et al., 2013), while it is decreased in the model projections (Cia et al., 2021). This supports our notion that the precipitation is a key indicator to enhance the NTA–, and SNP–ENSO relationship.”

Line 282: “El Niño-like mean state → the wetter mean state over the off-equatorial eastern Pacific”

Figure 2-D. (a) Area-averaged mean precipitation (mm·day⁻¹) over the off-equatorial eastern Pacific (150°–90° W, 2° S–8° N) during the spring-to-fall seasons (February to October) for 1976–1997 (*blue bar*) and 1998–2019 (*red bar*) in observations. (b) Same as (a), but for the 30 selected CMIP6 models under the present-day (1951–1999; *blue bar*) and future climate (2051–2099; *red bar*). Error bars are calculated as 1.0 s.d. of 10,000 inter-realizations of a bootstrap method.

Figure 2-E. The difference in the SST ($^{\circ}\text{C}$) climatology during 1998–2019 from that during 1976–1997 in ERSST v5. (b) Same as (a), but for the precipitation ($\text{mm}\cdot\text{day}^{-1}$) in ECMWF–ERA5.

Figure 2-F. (a) 21-yr moving correlation coefficients between the NTA index and the SNP index during the boreal spring season (FMA), and the subsequent year’s Niño3.4 index during the boreal winter season (DJF) (*blue line*: NTA vs. Niño3.4; *red line*: SNP vs. Niño3.4) from 1951–2020. The x-axis indicates the middle year in the 21-yr moving window (e.g., 1999 indicates the correlation coefficient from 1989–2009). The open circles represent correlation coefficients that are statistically significant at the 95% confidence level.

Minor comments

1. L30: differ → differs

[Reply] Corrected

2. L57: what 'intermediate climate model' is referring to.

[Reply] Sorry for the confusion about the terminology. In this study, the term “intermediate climate model” refers to a coupled model consisting of a Community Atmosphere Model version 3.1 (CAM3.1) and a Zebiak-Cane type 1.5-layer reduced-gravity ocean (RGO) model, known as the CAM3.1-RGO model (Zebiak and Cane, 1987; Clement et al., 1996). We have corrected the sentence as below:

Line 56-57: “an intermediate climate model consisting of a full atmospheric CGM coupled with a reduced-gravity ocean (RGO) model”

3. L93: 'Totally' → 'In total'

[Reply] Corrected

4. L223: 'improves' → 'is enhanced'

[Reply] Corrected

References

- Cai, W., Santoso, A., Collins, M., Dewitte, B., Karamperidou, C., Kug, J.-S., et al. (2021). Changing El Niño–Southern oscillation in a warming climate. *Nat. Rev. Earth Environ.*, 2(9), 628-644.
- Clement, A. C., Seager, R., Cane, M. A., & Zebiak, S. E. (1996). An ocean dynamical thermostat. *J. Clim.*, 9(9), 2190-2196.
- Fan, H., Yang, S., Wang, C., Wu, Y., & Zhang, G. (2022). Strengthening amplitude and impact of the Pacific Meridional Mode on ENSO in the warming climate depicted by CMIP6 models. *J. Clim.*, 35(15), 1-52.
- Ham, Y.-G., & Kug, J.-S. (2014). ENSO phase-locking to the boreal winter in CMIP3 and CMIP5 models. *Clim. Dynam.*, 43, 305-318.

- McKenna, S., Santoso, A., Gupta, A. S., Taschetto, A. S., & Cai, W. (2020). Indian Ocean Dipole in CMIP5 and CMIP6: characteristics, biases, and links to ENSO. *Scientific reports*, 10(1), 1-13.
- Medhaug, I., Stolpe, M. B., Fischer, E. M., & Knutti, R. (2017). Reconciling controversies about the 'global warming hiatus'. *Nature*, 545(7652), 41-47.
- Wang, L., Yu, J.-Y., & Paek, H. (2017). Enhanced biennial variability in the Pacific due to Atlantic capacitor effect. *Nature Commun.*, 8, 14887.
- Xia, F., Zuo, J., Sun, C., & Liu, A. (2023). The Atlantic Meridional Mode and Associated Wind–SST Relationship in the CMIP6 Models. *Atmosphere*, 14(2), 359.
- Zebiak, S. E., & Cane, M. A. (1987). A model el niñ–southern oscillation. *Mon. Wea. Rev.*, 115(10), 2262-2278.

REVIEWERS' COMMENTS

Reviewer #1 (Remarks to the Author):

I thank the authors for the material added to the manuscript, which nicely addresses my major concern. The causality between subtropical SST precursors and ENSO evolution is now better established, and the subsequent diagnostics (which all assume this causality) are now much more convincing. I thus can recommend acceptance of the manuscript for publication.

Reviewer #2 (Remarks to the Author):

Thank you to the authors who have made a significant effort to address the reviewers' questions and concerns. The new experiments are valuable in addressing the impacts of the NTA and SNP on ENSO. I am happy to recommend acceptance, if my comments below are addressed.

I appreciate the author's efforts to evaluate the phase locking of ENSO, NTA and SNP. While I agree that the phase locking is adequately represented in the CMIP6 models, it is clear from Supp Figs 1b and 2b that more than half of the CMIP5 models do not represent the NTA seasonal cycle accurately. In this case, I don't see any benefit in including CMIP5 models in your analysis.

As you note in your response and in your manuscript, CMIP6 models display an improvement in several areas.

My recommendation is to remove the CMIP5 models from the main figures & analysis. I note that in e.g. Figs 1, 2 & 6, you focus primarily on CMIP6 models anyway, so I don't see why CMIP5 models need to be included, especially if they don't simulate an accurate seasonal cycle.

However, the supplementary figures should be kept, and discussed, as they are useful in demonstrating the improvements in CMIP6.

L103-104: The statement that CMIP5/6 models accurately simulate the phase locking should be changed to reflect my point above. I would state that the simulation of the NTA seasonal cycle is poor CMIP5 and refer to the Supp Figs.

Minor comments:

L254: 'significantly is enhanced' -> 'is significantly enhanced'?

L282: 'significantly improve' -> 'significantly improves'

REVIEWER COMMENTS:

Reviewer #2:

Thank you to the authors who have made a significant effort to address the reviewers' questions and concerns. The new experiments are valuable in addressing the impacts of the NTA and SNP on ENSO. I am happy to recommend acceptance, if my comments below are addressed.

I appreciate the author's efforts to evaluate the phase locking of ENSO, NTA and SNP. While I agree that the phase locking is adequately represented in the CMIP6 models, it is clear from Supp Figs 1b and 2b that more than half of the CMIP5 models do not represent the NTA seasonal cycle accurately. In this case, I don't see any benefit in including CMIP5 models in your analysis.

As you note in your response and in your manuscript, CMIP6 models display an improvement in several areas.

My recommendation is to remove the CMIP5 models from the main figures & analysis. I note that in e.g. Figs 1, 2 & 6, you focus primarily on CMIP6 models anyway, so I don't see why CMIP5 models need to be included, especially if they don't simulate an accurate seasonal cycle. However, the supplementary figures should be kept, and discussed, as they are useful in demonstrating the improvements in CMIP6.

[Reply] Thank you for giving us useful comments. According to the reviewer's comments, we have removed the CMIP5 results from the main figures & analysis. With this modification, our results become much rigorous with high correlations between key components. However, we kept our original analysis by including CMIP5 for Figure 4b, and 4c, as the selected number of the models is very few with CMIP6 only; the number of selected number of models is 9, and 3 with CMIP5+6 results for Figure 4b, and 4c, while it is only 4, and 2 only with CMIP6 results, respectively. To ensure generality of our results, we decided to increase the number of models by using both CMIP5 and 6 model for Figure 4b and 4c. This information is clearly given in the figure caption.

Minor comments

L103-104: The statement that CMIP5/6 models accurately simulate the phase locking should be changed to reflect my point above. I would state that the simulation of the NTA seasonal cycle is poor CMIP5 and refer to the Supp Figs.

[Reply] We appreciate the reviewer's comment. To reflect the reviewer's suggestion, we added the following sentence in the revised manuscript:

Line 102-106: It should be noted that while the CMIP6 models accurately simulate the phase-

locking of ENSO, NTA, and SNP as the observed to some extent, while the simulation of the NTA seasonal cycle is relatively poor in the CMIP5 models (Figs. S1 and S2). Therefore, we focused on the CMIP6 model results to assess the changes in the role of NTA or SNP on the ENSO due to the global warming.

L254: 'significantly is enhanced' -> 'is significantly enhanced'?

[Reply] Corrected

2. L282: 'significantly improve' -> 'significantly improves'

[Reply] Corrected